# Exploring Structural Diversity among Adhesion Devices Encoded by Lactococcal P335 Phages with AlphaFold2

**DOI:** 10.3390/microorganisms10112278

**Published:** 2022-11-16

**Authors:** Adeline Goulet, Jennifer Mahony, Christian Cambillau, Douwe van Sinderen

**Affiliations:** 1Laboratoire d’Ingénierie des Systèmes Macromoléculaires (LISM), Institut de Microbiologie, Bioénergies et Biotechnologie (IM2B), Aix-Marseille Université—CNRS, UMR 7255, 13288 Marseille, France; 2School of Microbiology, University College Cork, T12 YN60 Cork, Ireland; 3AlphaGraphix, 24 Carrer d’Amont, 66210 Formiguères, France

**Keywords:** Bacteriophages, lactococcal P335 phages, phages adhesion device, phages structure, alphafold2

## Abstract

Bacteriophages, or phages, are the most abundant biological entities on Earth. They possess molecular nanodevices to package and store their genome, as well as to introduce it into the cytoplasm of their bacterial prey. Successful phage infection commences with specific recognition of, and adhesion to, a suitable host cell surface. Adhesion devices of siphophages infecting Gram-positive bacteria are very diverse and remain, for the majority, poorly understood. These assemblies often comprise long, flexible, and multi-domain proteins, which limit their structural analyses by experimental approaches. The protein structure prediction program AlphaFold2 is exquisitely adapted to unveil structural and functional details of such molecular machineries. Here, we present structure predictions of adhesion devices from siphophages belonging to the P335 group infecting *Lactococcus* spp., one of the most extensively applied lactic acid bacteria in dairy fermentations. The predictions of representative adhesion devices from types I-IV P335 phages illustrate their very diverse topology. Adhesion devices from types III and IV phages share a common topology with that of *Skunavirus* p2, with a receptor binding protein anchored to the virion by a distal tail protein loop. This suggests that they exhibit an activation mechanism similar to that of phage p2 prior to host binding.

## 1. Introduction

The lactococcal P335 phage quasi-species comprises a diverse group of phages that include temperate and virulent members, being among the most frequently encountered lactococcal phages in dairy fermentations [1]. These phages have been extensively studied with respect to their genetic diversity [1,2,3,4,5,6], host interactions [7,8], and their structural features [9,10,11]. The primary interaction of a phage with its cognate host is mediated by the so-called “adhesion device”, which is located at the distal end of the phage tail. The adhesion device is responsible for the recognition of, and binding to, a suitable host-encoded receptor. Many P335 phages exhibit an exceptionally narrow host range, which is believed to reflect a highly specific interaction with the host. The adhesion device of tailed phages typically comprises a multi-protein complex incorporating the distal tail protein (Dit), tail-associated lysin (Tal), and a receptor binding protein (RBP) with auxiliary adhesion proteins in some cases [12]. Detailed comparative genome and morphological analyses of P335 phages resulted in the identification of five adhesion device-associated groups (I-V) [2,13]. Members of group I (including BK5-T) typically possess long tail fibres, while group II phages (including TP901-1 and Tuc2009) exhibit bulky appendages [6]. Members of group III (e.g., r1t and LC3) and group IV (e.g., Q33 and BM13) possess “stubby” distal tail regions [6]. Group V phages appear to be a distinct group though are most closely related to the group I phages in relation to their adhesion device-encoding proteins [14].

The P335 phages Tuc2009 and TP901-1, which belong to group II, have become a paradigm for the interactions of these phages with their hosts and they are undoubtedly the best studied with respect to their structural characteristics [1,11,15,16]. While the adhesion devices of these phages are highly similar, a notable difference is the presence of the auxiliary binding protein, BppA, in Tuc2009 [11]. BppA was proven to enhance the (carbohydrate) binding capabilities of Tuc2009 although it is non-essential for infection [9]. Resolving the structure of BppA was transformative for understanding the interactions of Gram-positive infecting phages, many of which similarly recognize a carbohydrate receptor moiety. The BppA domain has been subsequently identified in multiple adhesion device proteins highlighting the widespread nature of such carbohydrate binding domains and their influence in host interactions [17,18,19,20]. The specificity of these phages is believed to be underpinned by the diverse nature of the cell wall polysaccharides (CWPS) that act as the receptor [21]. Furthermore, it has been demonstrated that two P335 phages (TP901-1 and phiLC3) that infect the same host strain follow a distinct DNA injection pathway [22]. It is noteworthy that these phages belong to distinct groups based on analysis of their adhesion devices and which may be linked to their receptor-binding, DNA-release, and infection processes. Therefore, it is imperative that we establish these intricate links between phage morphology/structure and host interactions in order to improve our ability to predict the receptor and/or host range of a given phage. While structural studies pertaining to TP901-1 have progressed significantly over the past decade, it is currently impossible to extrapolate the implications of these studies for other P335 phages (or indeed other lactococcal phage species).

The core of the adhesion devices of siphophages consists of an assembled Dit hexamer [23,24] bound to the last major tail protein hexamer (MTP), and a Tal trimer [25,26]. It is often completed by one or several receptor binding proteins (RBPs) [15,27,28,29,30]. Dit proteins can be divided into two domains corresponding to the N- and C-terminal parts of the polypeptide chain [23]. The N-terminal domain, called the belt, is composed of two β−sheets, a β−hairpin, and an α−helix. The C-terminal domain, named the galectin domain, is a two β−sheet structure, similar to a galectin domain. Of note, this galectin domain can be absent in some Dits, such as in phage Lambda, or can be replaced by an OB-fold domain, such as in phage T5 [24]. Dits possessing CBM insertions in the galectin domain are called “evolved Dits” [17,18,31,32]. In phage tails, six Dit monomers assemble as a ring allowing DNA passage. Tals are composed of an N-terminal structural domain of ~350–400 amino acids ([28,29] and unpublished structures in the PDB (3gs9, 3d37, 3cdd)). In many phages, this domain is followed by a C-terminal extension that is believed to play a role in peptidoglycan degradation, e.g., the *Lactococcus lactis* P335 phage TP901-1 [33] or host binding, e.g., the *B. subtilis* phage SPP1 [34] and *E. coli* phage T5 [35]. These extensions may encompass up to 1000–2000 residues, as in some *Streptococcus thermophilus* infecting phages [19]. Recently AlphaFold2 has transformed our ability to predict and analyse structures of phage adhesion devices, to compare them to each other and to make assertions and predictions for the mode of interaction of these phages with their hosts [36,37,38,39,40]. In the current study, we present a detailed analysis of the AlphaFold2 structural predictions of adhesion devices from P335 phages representing Groups I-IV, i.e., C41431 and BK5-T (Group I), TP901-1 and Tuc2009 (Group II), LC3 (Group III), and Q33 (Group IV) [2].

## 2. Materials and Methods

Protein Structure Predictions and Topological Model Assembly

A Github AlphaFold2 notebook (https://colab.research.google.com/github/deepmind/alphafold/blob/main/notebooks/AlphaFold.ipynb#scrollTo=XUo6foMQxwS2 (accessed on 31 October 2022)) was used to perform the predictions and the modelling of homo-multimers. Due to memory limitations, long sequences were split into sequence stretches with considerable overlap for later assembly. The number of residues in the multimeric stretch predictions had to be less than 1400 residues. In a first pass, structure predictions for monomers were prepared in order to determine applicable stretch boundaries to be assembled in trimers (Tal) or hexamers (Dit). Moreover, we predicted structures of stretches with overlapping segments to allow assembly of full-length multimers using the *Coot* superimposing options [41]. Local Distance Difference Test (LDDT) evaluates local distance differences of all atoms in a model with reference to an ensemble of equivalent structures. The pLDDT (predicted lDDT-Cα) is a per-residue measure of local confidence on a scale from 0–100 (100 being the highest confidence level). The pLDDT values that are stored in the pdb file as B-factors, were plotted using Excel (Appendix A). Visualizing the prediction quality along the amino-acid chain of the models provided as Appendix A can be achieved with display software such as *Coot* [41] or ChimeraX [42] using B-factors colouring mode. The best predicted parts are in red, the least in blue. The final predicted domain structures were submitted to the Dali server [43] to identify the closest structural homologs in the PDB. The Dali *Z* score is an optimised similarity score defined as the sum of equivalent residue-wise Cα–Cαdistances among two proteins. A higher value of *Z* score indicates greater similarity. The “lali” term is the number of aligned residue pairs.

In order to generate topological models of Dit-Tal assemblies, we used the *Coot* option “SSM Superpose” to superimpose individual domains onto the corresponding ones of the lactococcal phage p2 adhesion device [28]. These models do not result from AlphaFold2 predictions and should be considered as illustrations of possible topologies. Sequence alignments were performed with Multalin [44] and ESPript [45]. Visual representations of the structures were prepared with ChimeraX [42]. Analyses of protein–protein interfaces were performed using the PDBePISA server [46].

## 3. Results

### 3.1. Phages Belonging to P335 Type I: C41431 and BK5-T

We selected two P335 phages belonging to type I, C41431 and BK5-T, based on their Tal diversity. Both phages possess evolved Dits encompassing 512 and 548 amino acids, respectively, while their Tal proteins, consisting of 857 and 1904 amino acids, respectively, are quite different (Figure 1A,B). C41431 Dit contains a carbohydrate-binding module (CBM) insertion in its galectin domain’s loop 177-487 (Figure 1A Dit). The Dali server revealed that this insertion is a CBM which resembles the CBM present in *Lactobacillus casei* Dit (PDB-ID: 5ly8; Z-value: 19.9; rmsd:2.8Å; lali: 201/230 amino acids; [32]). Additionally, the BK5-T Dit has a CBM inserted in its loop 177-487 (Figure 1B Dit); Dali predicts that this CBM is closely related to the CBM of *L. casei* Dit (PDB-ID: 5ly8; Z-value: 18.7; rmsd:2.8Å; lali: 192/230 amino acids [32]).

Phage C41431 Tal is predicted to be 45 nm long. It exhibits an N-terminal domain resembling that of coliphage T4 gp27, phage p2 Tal [28] and several other phage Tal domains (Figure 1). This domain is followed by a helical trimer located along its 3-fold axis and abutting to a trimer of an immunoglobulin-like domain (PDB-ID: 1j86; Z-value: 9.9; rmsd: 1.8Å; lali: 77/173 amino acids). It is followed by a short trimeric collagen-like linker and a so-called 3β domain previously observed in streptococcal phages [20]. Another short collagen linker is followed by a large well-structured trimeric domain encompassing four sub-domains: a 3 × 4 β-stranded domain, a 3 × 3 β-stranded β-helix, a 3 × 11 β-stranded β-prism, and a C-terminal β-stranded domain. Dali reported hits of this latter domain with the RBP head domain (the receptor binding domain, RBD) of lactococcal phages bIL170 (PDB-ID: 2fsd; Z-value: 13.8; rmsd: 1.9Å; lali: 101/110 amino acids; [47]) and p2 (PDB-ID: 1zru; Z-value: 11.0; rmsd: 2.3Å; lali: 99/263 amino acids; [27]), thus identifying it as the carbohydrate-binding domain of phage C41431 (Figure 1A).

Phage BK5-T Tal is 140 nm long (Figure 1B). It also exhibits a T4 gp27-like N-terminal domain followed by a helical trimer located along its 3-fold axis and abutting to a trimer of an Ig-like domain similar to that of C41431. In contrast with phage C41431, two large domains in tandem follow these Ig-like domains. The first domain (residues 541–653) is a β-sandwich of 4 + 5 β-strands. Dali reported hits with various glycosyl hydrolases (GHs), the best hit being a CBM recognizing α-1,4-glucan from *Paenibacillaus* α-1,6-GH31 (PDB-ID: 5x7o; Z-value: 9.8; rmsd: 2.6Å; lali: 108/1247amino acids; [48]). The second domain (residues 669-794) is also a 4+4 β-sandwich, reported by Dali as similar to a CBM (Cazy CBM70) from *Streptococcus pneumoniae* hyaluronate lyase (PDB-ID: 4d0q; Z-value: 8.2; rmsd: 2.7Å; lali: 100/161 amino acids; [49]). It is followed by a short collagen-like linker and a 3β domain as observed in C41431 Tal, but repeated 10 times, thereby providing the huge extension of BK5t Tal. A large domain (residues 1639–1904) resembling that of C41431 follows, associating also a trimeric β-stranded domain, a β-helix, a β-prism and a C-terminal β-stranded domain. In BK5-T, Dali reported a similarity between this latter sub-domain and the receptor binding domain (RBD) of *L. lactis* phage TP901-1 RBP (PDB-ID: 4ios; Z-value: 18.0; rmsd: 1.1Å; lali: 100/100 amino acids; [50]) (Figure 1B).

### 3.2. Phages Belonging to P335 Type II: TP901-1 and Tuc2009

The adhesion device structure of lactococcal phage TP901-1, including Dit, BppU, and RBP has previously been determined by X-ray crystallography [15]. However, Tal was missing in this structure since the complete adhesion device complex could not be produced [15,51]. We therefore predicted the structure of TP901-1′s Tal (residues 1-918) (Figure 2A). As in all Tal proteins, the TP901-1 Tal N-terminus has a T4 phage gp27 fold [52] and resembles that of p2 phage [28] or streptococcal phages [20]. As with type I phages, three helices fill and extend outside the trimer centre (Figure 2A). They are followed by three structural domains (residues 401–589): a 3 × 5 β-stranded β-prism, a 3 × 8 large β-stranded β-prism, and a 4-turn β-helix. This elongated Tal central part is linked to two domains (residues 618–918) by a long linker (Figure 2B). The first domain (residues 618-772, Figure 2C) is reported by Dali as resembling a domain of a cell-wall-degrading enzyme of the *Bacillus* bacteriophage phi29 (PDB-ID: 3csq; Z-value: 20.7; rmsd: 1.7Å; lali: 143/324 amino acids; [53]). This domain is structurally related to lysozymes and cleaves intersaccharidic bonds of the peptidoglycan. The phi29 domain possesses two residues involved in polysaccharide binding and cleavage, Glu45 and Asn54 (Asn55 in the PDB file) [53]. In the first TP901-1 domain, the corresponding residues are Glu651 and Glu660.

The second domain (782–918; Figure 2D) is reported by Dali as being close to a lysostaphin metalloendopeptidase (LytM) able to cleave a Gly-Gly bond in a tetraglycine motif bridging peptidoglycan (PDB-ID: 2b13; Z-value: 19.9; rmsd: 1.6Å; lali: 125/131 amino acids; [54]). To note, the second of the cell-wall-degrading enzymes from *Bacillus* bacteriophage Phi29 shares the same fold, although more remote compared to the first hit (PDB-ID: 3csq Z-value: 13.5; rmsd: 2.3Å; lali: 122/326 amino acids; [53]). The three domains share superimposable Zn binding residues, His210, His 293, Asp 214 for LytM (2b13), His188, His280, Asp195 for the Phi29 enzyme, and His 810, H894 and Asp814 for TP901-1 Tal.

We assembled TP901-1 Tal within the Dit/BppU/RBP crystal structure, using the lactococcal phage p2 adhesion device crystal structure as the template (Figure 2E–H). In the complete TP901-1 adhesion device hybrid model, Tal protrudes by ~150 Å out of the 150 Å-thick Dit/BppU/RBP complex, which is in agreement with the negative staining electron microscopy images of P335 phages [2] (Figure 2E,F).

Tuc2009 and TP901-1 baseplate components exhibit significant amino acid conservation. Their Dits and Tals display ~96% identity (with an 11-residue insertion in Tuc2009 Tal), while their BppUs exhibit sequence disparity at the C-termini. Their RBPs also display distinct sequences (Appendix A). Tuc2009 BppU differs from TP901-1 BppU as it binds to a supplementary component, BppA, which is absent in TP901-1 [11,15]. Similarly, the C-terminal domain of their RBPs differ in sequence and exhibit different folds. We assembled the complete adhesion device of phage Tuc2009 (Figure 3). The 12 BppA module at the baseplate periphery provides a significantly enhanced, although not essential, affinity for the Tuc2009 host [9].

### 3.3. Phages Belonging to P335 Type III and IV: PhiLC3 and Q33

The adhesion device of phage phiLC3 (type III) assembles three components: Dit, Tal, and RBP. We predicted the structures of Dit, Tal, and RBP separately. The RBP trimer of PhiLC3 possesses three domains, resembling those of phage p2 [28], from N- to C-terminus: a trimeric β-sandwich domain termed the shoulders, an α-helical stretch followed by a 4-turn β-helix domain termed the neck, and a trimeric C-terminal β-sandwich domain called the head (Figure 4A). The shoulder domain exhibits a domain-swapping exchange of its 15 N-terminal residues, forming a triangular cavity at the bottom of the domain. The trimeric C-terminal head domain is roughly reminiscent of the RBP head domains of lactococcal phages p2, TP901-1, Tuc2009 and Listeria phage PSA [11,27,30,50]. The sequence identity between phiLC3 RBP head domain and that of TP901-1 is 95%, with conserved receptor-binding residues (Appendix A) reflecting the fact that TP901-1 and phiLC3 are able to infect the same lactococcal strain (i.e., *L. cremoris* 3107). Indeed, Dali reported a strong hit with the TP901-1 RBP head (PDB-ID: 4ios; Z-value: 21.2; rmsd: 0.6 Å; lali: 100/100 amino acids; [50]). PhiLC3 Dit possesses the classical Dit modules, belt, and galectin, and does not harbour a CBM insertion. Instead, as is the case for phage p2, it exhibits an insertion of two antiparallel β-strands with a triangular-shaped loop between them, together called “arm and hand” in phage p2 Dit [28] (Figure 4B and Appendix A). We submitted the RBP sequence (three times repeated) together with the complete Dit sequence to AlphaFold2 for multimer structure prediction. The predicted structure of the complex exhibits an excellent fit between the RBP central shoulder cavity and the Dit hand loop (Figure 4B,D,E). The pLDDT of the RBP N-terminus and the Dit arm and hand domain exhibits much higher values in the complex compared to the isolated proteins (Appendix A).

Superposition of the empty RBP shoulder domain with the shoulders domain in the complex revealed that the N-terminus (residues 6–20) of each RBP subunit has moved upon complexation in order to accommodate Dit’s hand domain (Figure 4D). This interaction covers 1450 Å^2^ of the RBP surface and 1410 Å^2^ of the Dit surface. This interaction is established through two kinds of interactions including (i) several hydrogen bonds between mainchain C=O and N-H moieties of RBP residues 6–22 and the arm loop residues 158–173 (Appendix A), and (ii) hydrophobic/aromatic interactions (Figure 4D and Appendix A).

Finally, the Tal is restrained to its N-structural domain, resembling that of phage p2 in its closed conformation [28]. Assuming that the individual adhesion device components are structurally similar to those of the phage p2 adhesion device, we assembled a complete topological model using Coot [41] of the phage p2 adhesion device in its resting conformation (PDB ID 2wzp) as template (Figure 5 and Appendix A). Note that in phage p2 the baseplate displays two conformations, one being the resting state, with the RBPs pointing to the top, the second being the activated form, with the RBPs pointing to the bottom (see discussion).

The adhesion device of phage Q33 also consists of three components: Dit, Tal and RBP, of which we predicted the structures separately. The Dit also possesses an arm and hand extension, the Tal is short and non-evolved and the RBP trimer of Q33 possesses three domains (with a long neck), and all of these features are similarly observed in phiLC3 (Figure 6A). As in phiLC3, the Dit hand fits exquisitely into the shoulder cavity (Figure 6B,C). Dali reported that the Q33 RBD is very similar to that of Tuc2009 (PDB-ID: 5e7f; Z-value: 20.6; rmsd: 1.1Å; lali: 114/128 amino acids; [11]). Here, also, we manually assembled the whole Q33 adhesion device using the phage p2 adhesion device structure in its resting state as template (Figure 6D).

### 3.4. The Neck Passage Structure of Phages TP901-1, Tuc2009 and PhiLC3

The three phages TP901-1, Tuc2009, and phiLC3 contain a long-predicted gene encoding a ~700 amino acid long protein, downstream of the gene encoding their RBP and exhibit an extended structure branched on their collar at the capsid-tail junction. Such a structure, named the neck passage structure (NPS) [17,55], is believed to serve as an ancillary phage adsorption device, aimed at capturing the phage’s host in an initial, yet reversible contact.

The sequences of the three ORFs are very similar between residues 1–440 and 588–700 (Figure 7A). Between these conserved regions, the sequences substantially differ and are poorly aligned. We predicted the structure of the three NPS proteins (Figure 7b). Their N-terminus (1–120) is a β-sandwich domain resembling that of the N-terminus of the BppU adaptor protein in phages TP901-1 and Tuc2009 (PDB id: 3uh8; Z-value: 11.6; rmsd; 3.1; lali 112/118 amino acids, [15]), which has been shown to anchor onto the Dit hexamer. This domain is followed by a long, irregular, coiled-coiled helical trimer (residues 122-365) and a trimeric β-stranded domain identified by Dali as CBMs (Figure 7B). A linker projects each CBM out of the 3-fold axis. Each CBM is followed by another linker and a loop, and the NPS protein terminates by trimeric β-hairpins forming a kind of umbrella over the CBM trimer (Figure 7B).

As mentioned above, the sequence of these three NPS-associated CBMs are very different. Accordingly, their structures differ significantly. For TP901-1, Dali retrieved a remotely similar CBM (CBM15 in Cazy nomenclature [55]), a xylan-binding module from a *Pseudomonas cellulosa* xylanase (PDB id: 1gny; Z-value: 7.2; rmsd: 3.1 Å; lali 111/153 amino acids; [56]). Dali identified a better hit for the Tuc2009 NPS CBM (Cazy CBM61), a CBM component of *Thermotoga maritima* α-1,4-galactanase (PDB id: 2xon; Z-value: 11.8; rmsd: 2.6 Å; lali 124/145 amino acids; [57]). Finally, Dali reported a hit for the PhiLC3 CBM with a CBM component (Cazy CBM4-2) of *Rhodothermus marinus* xylanase (PDB id: 2y6h; Z-value: 14.2; rmsd: 2.2 Å; lali 137/167 amino acids; [58]). Of note, despite a common host and identical RBPs in phages TP901-1 and PhiLC3, the CBMs of their NPS differ substantially.

## 4. Discussion

Although belonging to the same P355 group of lactococcal phages, the adhesion devices of the six phages examined in this study exhibit tremendous differences (Appendix A). The two phages from type I, C41431 and BK5-T, possess comparable domains but differ in two aspects. The first difference is that the 3β-domain which is present as one unit in C41431 is repeated ten times in BK5-T, providing a Tal extension of 140 nm instead of 45 nm. The second difference is the presence in BK5-T of two CBM domains in tandem following the Ig-like domain. In both phages, a Tal C-terminal domain resembles the trimeric RBP head of phages p2 [27] and TP901-1 [50], respectively. Hence, C41431 harbours three adhesion sites on its RBP compared to BK5-T which harbours nine of them. In any case, these numbers of adhesion sites are lower than those observed in phage p2 (18 sites) [28] or TP901-1 (54 sites) [15]. This may be compensated by the presence of six adhesion sites in the Dit of each phage. Of note, the BK5-T’s tail tip architecture resembles that of *S. thermophilus* phages [20] and of *L. casei* phage J1 which also possess a long Tal with six 3β-domains and a terminal RBD [40]. J1 Dit is also evolved and possesses CBMs resembling those of C41431 and BK5-T. Most probably, the evolved Dit plays an essential role in the three phages by locking them on the host’s cell wall, while the Tal may serve to perform a preliminary exploration of possible preys.

The architecture of the adhesion devices of type II phages, TP901-1 and Tuc2009, has been extensively documented [9,11,26]. However, the structure of their Tals was never reported previously, preventing a complete structural analysis of their adhesion devices. In contrast with the elongated Tals of type I phages, Tals of type II phages are very large and formed of parallel β-strands roughly perpendicular to the trimer axis. These Tals are 210 Å long and extend beyond the RBP bundle. Another difference with the type I Tals is that they do not harbour adhesion domains but exhibit two enzymatic domains at their C-terminus. The proximal domain is a peptidoglycan polysaccharide-degrading enzyme, while the distal domain is similar to tetra-Gly endopeptidases. This tandem topology is also present in the *Bacillus* phage Phi29 enzyme [53]. However, such tetra-Gly motif is not observed in the *Bacillus* and *Lactococcus* strains infected by Phi29 and TP901-1 [33,53,56], suggesting that both enzymes may have a broader or different specificity. A tetraglycine motif is present in the linker between the two C-terminal catalytic domains between Gly778 and Gly781 (S**GGGG**Y). Another Gly rich motif is located before the cell-wall-degrading enzyme, between Gly597 and Gly604 (N**GGG**NS**GGGD**) [2,33]. It was originally suggested that the endopeptidase is involved in Tal autocleavage [56]. However, disrupting the catalytic domain of the endopeptidase by mutating one of the Zinc-binding histidine did not abolish Tal cleavage, suggesting that Tal processing occurs through an as of yet unknown mechanism [33].

Adhesion devices of type III and IV phages reveal high similarity and remarkable features. They are formed of classical hexameric Dits, each exhibiting arm and hand extensions observed in phage p2, short Tals comprising only the T4 gp27 N-terminal domain [52], and trimeric RBPs resembling that of phage p2 with conserved shoulder, neck, and head domains [27]. In contrast with phage p2, the neck is much longer and assembles a trimer of long helices and a β-helix resembling that of phage p2. In phage p2, the hand triangular loop fits within a triangular cavity formed in the shoulder trimer. Strikingly, this type of assembly has been predicted by AlphaFold2, revealing strong molecular and structural complementarity between the two partners, and leading to an overall architecture resembling that of phage p2. Such an architecture has previously been proposed, based on AlphaFold2 predictions, for the *Oenococcus oeni* phage OE33PA [25]. The functional consequence of these adhesion device architectures is that they should be subjected to the same activation mechanism as that observed in phage p2 in order to orientate the RBPs in a position more favourable for host adhesion [28] (Appendix A). Hence, phages with an “articulated” adhesion device may be much more widespread than initially thought.

On top of the tail tip adhesion devices, phages TP901-1, Tuc2009 and PhiLC3 possess extra adhesion modules, the NPS, attached to phages’ collars. Such modules have been previously analysed in TP901-1 and in the *Skunavirus* species (previously known as the 936 group) [17]. Of note, these elongated structures belong to phages that possess short Tals or that do not incorporate CBMs in their Tals. Therefore, they may significantly contribute to the initial capture of prey in the phage’s environment.

Although phage tail tips deeply vary from one another, conserved domains and domain arrangements seem to be scrambled between them. Based on our recent AlphaFold2 studies of phage adhesion devices and with more results to come, it may be envisaged that we will soon understand the functional rationale of such amazing combinations in terms of their contribution to infection fitness.

## Figures and Tables

**Figure 1 microorganisms-10-02278-f001:**
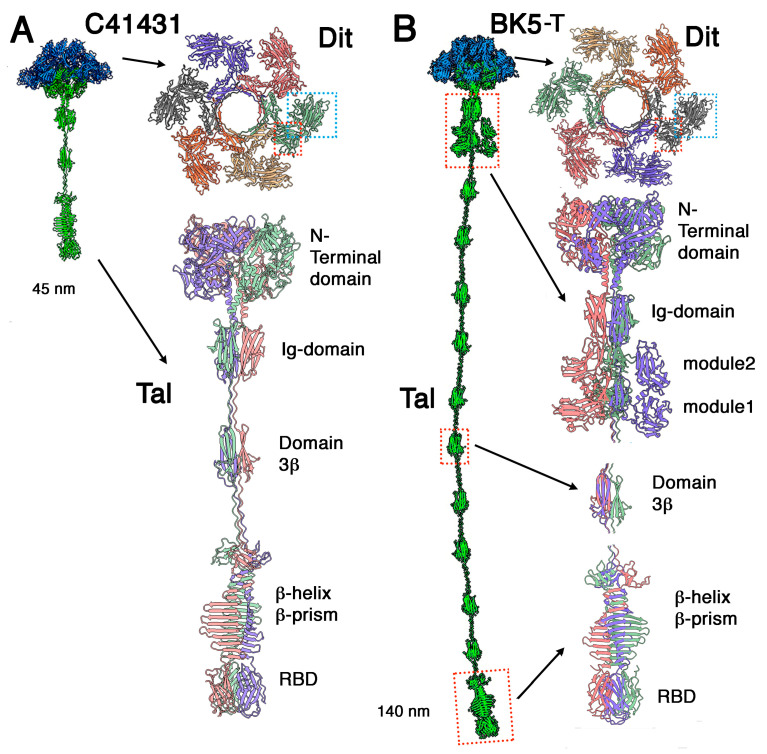
**Ribbon representations of predicted structures of adhesion devices from P335 type I phages.** (**A**) Phage C41431. Left: Topological model of the whole adhesion device with the Dit hexamer in blue and the Tal trimer in green. Top right: the evolved Dit hexamer (coloured by chain) with one of its CBM highlighted with a blue dotted box. Bottom right: The trimeric Tal (coloured by chain) with its domains joined by collagen-like structures. (**B**) Phage BK5-T. Left: Topological model of the whole adhesion device with the Dit hexamer in blue and the Tal trimer in green. Top right: the evolved Dit hexamer (coloured by chain) with one of its CBM highlighted with a blue dotted box. Bottom right: close-up views on the domains highlighted with red dotted boxes in the topological model.

**Figure 2 microorganisms-10-02278-f002:**
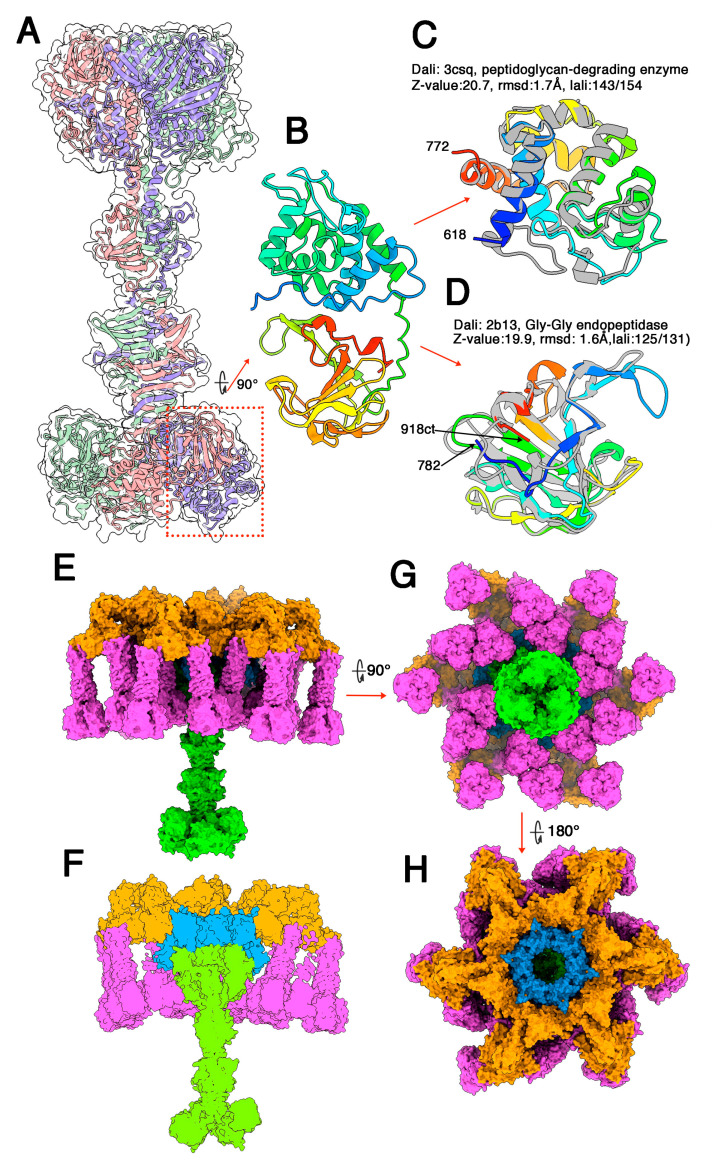
**Topological model of the phage TP901-1 (P335 type II) adhesion device**. (**A**) Ribbon representation of the trimeric Tal (coloured by chain) and its surface (transparent). The two catalytic domains are boxed with red dots. (**B**) View of the two catalytic domains (rainbow coloured) rotated by 90º as compared with (**A**). (**C**) Ribbon representations of the peptidoglycan-degrading enzyme (rainbow coloured) superimposed into its Dali closest hit (grey). (**D**) Ribbon representation of the Gly-Gly endopeptidase (rainbow coloured) superimposed into its Dali closest hit (grey). (**E**) Surface side view of the complete adhesion device (Dit: blue; BppU: orange; RBP: magenta; Tal: green). (**F**) Slabbed view of the complete adhesion device showing contacts between the different components (same colours as in (**E**)). (**G**) Surface bottom view of the complete adhesion device (same colours as in (**E**)). (**H**) Surface top view of the complete adhesion device (same colours as in (**E**)).

**Figure 3 microorganisms-10-02278-f003:**
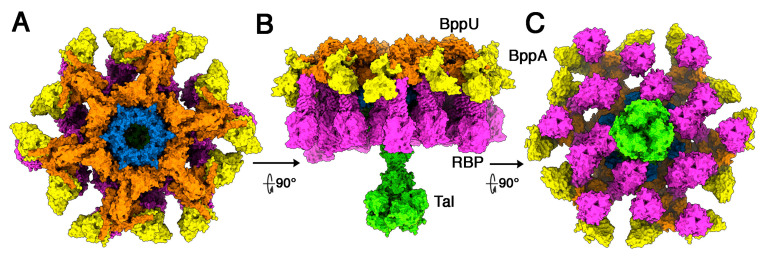
**Topological model of the phage Tuc2009 (P335 type II) adhesion device**. (**A**) Surface top view of the complete adhesion device (Dit: blue; BppU: orange; BppA: yellow; RBP: magenta; Tal: green). (**B**) Surface side view of the complete baseplate model (same colours as in (**A**)). (**C**) Surface bottom view of the complete baseplate model (same colours as in (**A**)).

**Figure 4 microorganisms-10-02278-f004:**
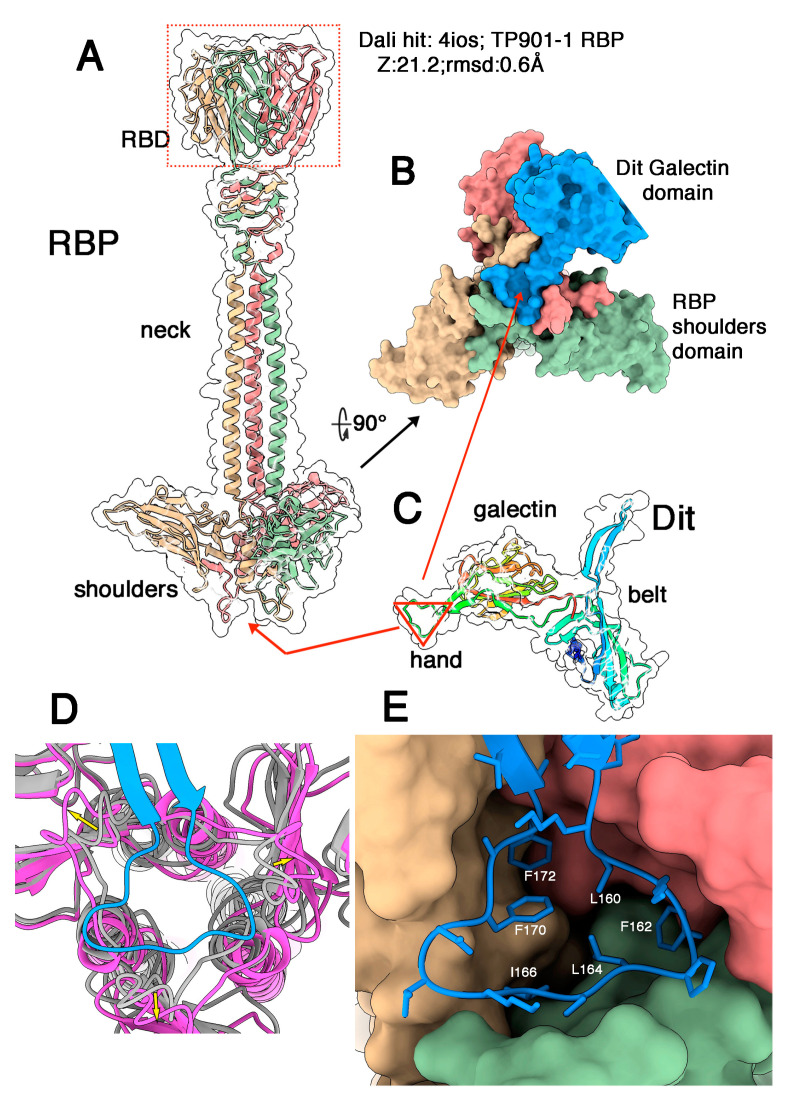
**The Dit/RBP interaction in phage phiLC3.** (**A**) Ribbon representation of the predicted structure of the RBP trimer (coloured by chain) with its three domains analogous to those of phage p2 RBP, and its transparent surface. The RBDs are highlighted in a red dotted box. (**B**) Surface representation of the RBP shoulder domains (same colour as in (**A**)) with the interacting Dit galectin, arm and hand domains (blue). (**C**) Ribbon representation of a Dit monomer (rainbow colour) and its transparent surface. The hand domain is identified by a red triangle. (**D**) Close-up view of the Dit arm domain (blue) interacting with the RBP shoulder domains (magenta), superimposed to the RBP shoulders in the absence of the Dit arm domain (grey). (**E**) Close-up view of the Dit arm domain (blue ribbon) interacting with the RBP shoulder domains (surface coloured by chain) with the Dit-interacting residues shown as sticks.

**Figure 5 microorganisms-10-02278-f005:**
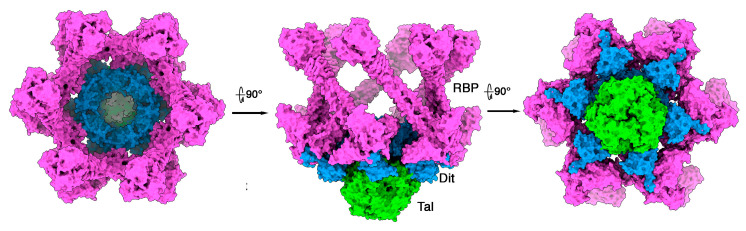
**Topological model of the phage phiLC3 (P335 type III) adhesion device**. Surface representations of the complete adhesion device (Dit in blue, RBP in magenta, and Tal in green). Top (left), side (middle) and bottom (right) views are shown.

**Figure 6 microorganisms-10-02278-f006:**
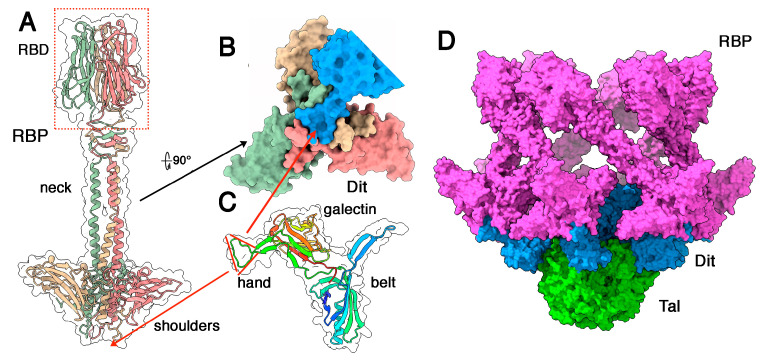
**The Dit/RBP interaction in Q33 and the adhesion device topological model.** (**A**) Predicted structure of the RBP trimer (coloured by chain) with its three domains structurally close to those of phage p2 RBP, and its transparent surface. The RBDs are highlighted in a red dotted box. (**B**) Surface representation of the RBP shoulders (same colour as in (**A**)) with the interacting Dit galectin, arm, and hand domains (blue). (**C**) Ribbon representation of a Dit monomer (rainbow colour) and its transparent surface. The hand domain is identified by a red triangle. (**D**) Surface side view of the complete adhesion device (Dit in blue, RBP in magenta, and Tal in green).

**Figure 7 microorganisms-10-02278-f007:**
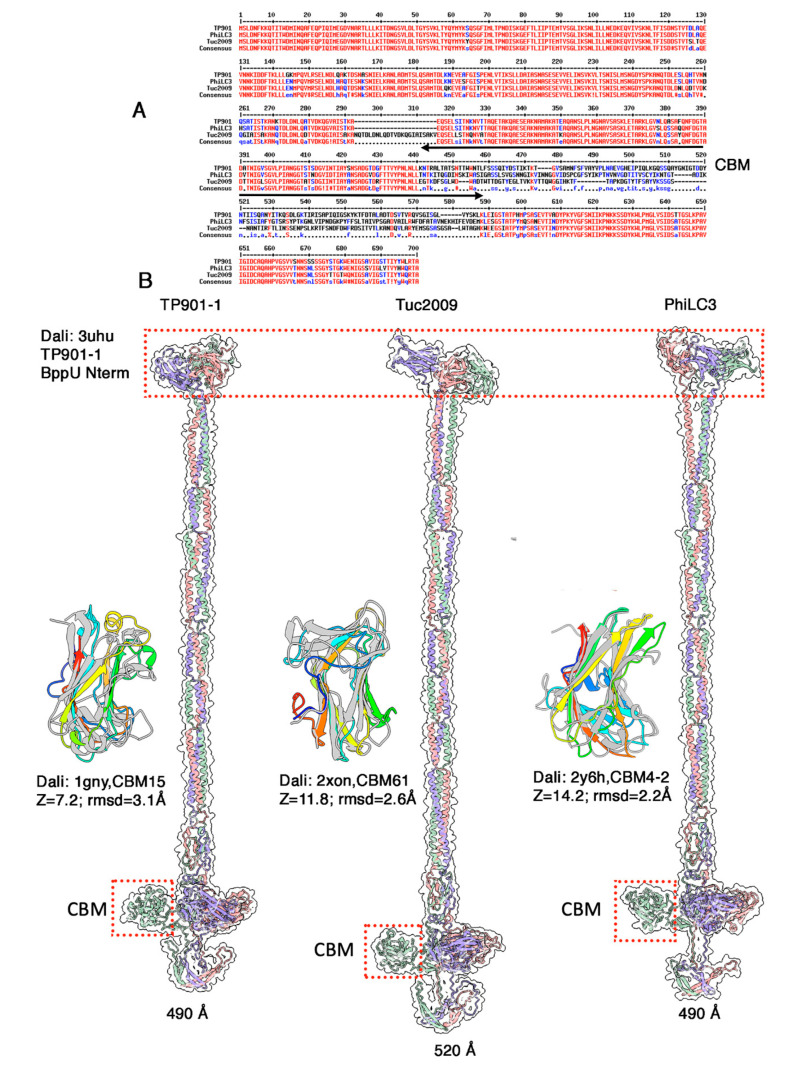
The NPS protein from phages TP901-1, Tuc2009 and PhiLC3. (**A**) Sequence alignment of the three NPS proteins. Note that sequences diverge in the CBM region. (**B**) Ribbon representation of the three predicted structures (coloured by chain) with transparent surfaces. The BppU domains are highlighted in a single red dotted box. One CBM of each protein is highlighted in a red dotted box. These CBMs (rainbow coloured) superimposed into their Dali’s best hit (grey) are shown on the left of each trimer.

## Data Availability

The coordinates of the AlphaFold2 predicted structures are available as Appendix A.

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
