# Peer review of "Exploring Structural Diversity among Adhesion Devices Encoded by Lactococcal P335 Phages with AlphaFold2"

_microorganisms, 2022, doi:10.3390/microorganisms10112278_

Round 1
Reviewer 1 Report
In their paper, Goulet and collaborators analyse the predicted structure of the host adsorption device of bacteriophages, belonging to different types of the P335 Lactococcus spp. group. With the revolution of the AlphaFold2 software in structural biology, these authors systematically investigate the host adsorption device of phages infecting different Gram+ bacteria (Oenococcus oeni, Streptococcus thermophilus and other Streptococcus phages, J1).
The data is clearly presented and well-illustrated. The authors cite the literature adequately, and give credit to related research.
I would however have a few comments:
- An important tool used in the paper is structural homology searches using the Dali software. For the readers not familiar with reading the results, a quick explanation of the output would be helpful (Z-value? lali?)
- The authors investigate the host adsorption devices of different phages, and propose long to very long central fibres. Are those indeed visible on negative stain images of these phages? Are the lengths compatible with the predicted structure?
- Line 239: it is indicated that the Dit galectin domain, together with the arm and the hand were fed into AlphaFold2 with the RBP sequence to predict the structure of the complex. However, in Fig. S1B, the pLDDT spans the whole Dit sequence. This looks inconsistent.
- The pLDDT of the predicted structures, which gives an indication on the confidence of the prediction, should be more commented. For example, the predicted structures could be showed coloured by the pLDDT value. This would indicate which domains are more poorly predicted. Presumably the linkers between domains?
- It is not clear from the text (lines 271-279) that the “exquisite fit” also needed the structure of the complex predicted rather than the individual proteins. Reading the paragraph, I understand predicted structures have been docked, Figure S1B suggests rather that the complex has been predicted.
- Fig. 7, labelling the CBM domain would help.
- For phages PhiLC3 and Q33, it is very intriguing to see the RBD pointing upwards rather that downwards, towards the surface of the host bacterium. A cryptic phrase alludes to an activation mechanism similar to that of p2. Details could be welcome!
- Legend to figure 1: prefer “top” and “bottom” rather than “up” and “down”?
Author Response
We would like to thank the reviewers for their comments and their suggestions to improve the manuscript. Below are our answers to the questions they raised.
Reviewer 1
In their paper, Goulet and collaborators analyse the predicted structure of the host adsorption device of bacteriophages, belonging to different types of the P335 Lactococcus spp. group. With the revolution of the AlphaFold2 software in structural biology, these authors systematically investigate the host adsorption device of phages infecting different Gram+ bacteria (Oenococcus oeni, Streptococcus thermophilus and other Streptococcus phages, J1).
The data is clearly presented and well-illustrated. The authors cite the literature adequately, and give credit to related research.
I would however have a few comments:
- An important tool used in the paper is structural homology searches using the Dali software. For the readers not familiar with reading the results, a quick explanation of the output would be helpful (Z-value? lali?)
Answer: we added in the methods section: " The Dali Z score is an optimised similarity score defined as the sum of equivalent residue-wise C α -C α distances among two proteins. A higher value of Z score indicates greater similarity. The "lali" term is the number of aligned residue pairs.
- The authors investigate the host adsorption devices of different phages, and propose long to very long central fibres. Are those indeed visible on negative stain images of these phages? Are the lengths compatible with the predicted structure?
Answer: we have added a supplementary figure (Supp. Fig. 7) in which we compare the Tal/baseplate models with single images of phages from each of the four types. However, concerning the type 1, we did not find pictures of long Tals.
- Line 239: it is indicated that the Dit galectin domain, together with the arm and the hand were fed into AlphaFold2 with the RBP sequence to predict the structure of the complex. However, in Fig. S1B, the pLDDT spans the whole Dit sequence. This looks inconsistent.
Answer: This was clearly a mistake; we first used the galectin, arm and hand domains and subsequently used the full Dit. We have changed the sentence to "We submitted the RBP trimer together with the complete Dit to AlphaFold2 for multimer structure prediction"
- The pLDDT of the predicted structures, which gives an indication on the confidence of the prediction, should be more commented. For example, the predicted structures could be showed coloured by the pLDDT value. This would indicate which domains are more poorly predicted. Presumably the linkers between domains?
Answer: Indeed the linkers are poorly predicted. However, providing images of each domain with colored pLDDT values is not really useful as we provide the PDB structures as supplementary data. People interested to look at details of the structure could use display programmes such as Pymol and ChimeraX and display the pLDDT values using the B-factor coloring option, as pLDDT are stored in the B-factors column. We have added the following sentence in the Materials and Methods section: "Visualizing the prediction quality along the amino-adic chain of the models provided as supplementary data can be achieved with display softwares such as Coot [41] or ChimeraX [42] using B-factors coloring mode. The best predicted parts are in red, the least in blue. "
- It is not clear from the text (lines 271-279) that the “exquisite fit” also needed the structure of the complex predicted rather than the individual proteins. Reading the paragraph, I understand predicted structures have been docked, Figure S1B suggests rather that the complex has been predicted.
Answer: The RBP/Dit complexes where obtain by submitting 3 RBP sequences and a Dit sequence to AlphaFold2 multimer. Hence, the structure of the complex is predicted as a whole. Besides, the structures of the RBP trimer and the Dit where predicted separately, allowing thus a comparison of the structures alone and in the complex (see Fig. 4E). We made in clearer in the text: "We submitted the RBP sequence (3 times repeated) together with the complete Dit sequence to AlphaFold2 for multimer structure prediction. The predicted structure of the complex exhibits an excellent fit between the RBP central shoulder cavity and the Dit hand loop (Fig. 4B,D,E)."
- Fig. 7, labelling the CBM domain would help.
Answer: CBMs have been labelled in Figure 7
- For phages PhiLC3 and Q33, it is very intriguing to see the RBD pointing upwards rather that downwards, towards the surface of the host bacterium. A cryptic phrase alludes to an activation mechanism similar to that of p2. Details could be welcome!
Answer: In phage p2, the baseplate has been observed in two conformations, one being the resting state, with the RBPs pointing to the top, the second being the activated form, with the RBPs pointing to the bottom. We modelled only the resting state. We assume that, as in phage p2, an activation mechanism should occur to orientate RBPs in a favourable position for host binding (see Supp. Fig 4).
We modified the sentence in results: "Assuming that the individual adhesion device components are structuraly similar to those of the phage p2 adhesion device, we assembled a complete topological model using Coot [41] of the phage p2 adhesion device in its resting conformation (PDB ID 2wzp) as template (Fig. 5; Supp. Fig. S4B-D). Note that in phage p2 the baseplate displays two conformations, one being the resting state, with the RBPs pointing to the top, the second being the activated form, with the RBPs pointing to the bottom (see discussion). "
And in discussion:" The functional consequence of these adhesion device architectures is that they should be subjected to the same activation mechanism as that observed in phage p2 in order to orientate the RBPs in a position more favourable for host adhesion [28] (Supp. Fig. S4 C,E)."
- Legend to figure 1: prefer “top” and “bottom” rather than “up” and “down”?
Answer: corrected
Figure 1. Ribbon representations of predicted structures of adhesion devices from P335 type I phages. (A) Phage C41431. Left: Topological model of the whole adhesion device with the Dit hexamer in blue and the Tal trimer in green. Top right: the evolved Dit hexamer (colored by chain) with one of its CBM highlighted with a blue dotted box. Bottom right: The trimeric Tal (colored by chain) with its domains joined by collagen-like structures. (B) Phage BK5-T. Left: Topological model of the whole adhesion device with the Dit hexamer in blue and the Tal trimer in green. Top right: the evolved Dit hexamer (colored by chain) with one of its CBM highlighted with a blue dotted box. Bottom right: close-up views on the domains highlighted with red dotted boxes in the topological model
Reviewer 2 Report
I read with great pleasure the manuscript and did not find anything to comment on, except two minor points.
L75. and an alpha-helix
L289. instead of "a long ORF ..." I would use "a long predicted gene encoding a ~700 amino acid long protein, downstream ..."
Author Response
We would like to thank the reviewers for their comments and their suggestions to improve the manuscript. Below are our answers to the questions they raised.
Reviewer 2
I read with great pleasure the manuscript and did not find anything to comment on, except two minor points.
L75. and an alpha-helix
Answer: corrected
L289. instead of "a long ORF ..." I would use "a long predicted gene encoding a ~700 amino acid long protein, downstream ..."
Answer: corrected